# A Survey of Genome-Wide Genetic Characterizations of Crossbred Dairy Cattle in Local Farms in Cambodia

**DOI:** 10.3390/ani12162072

**Published:** 2022-08-14

**Authors:** Somony Mam, Bengthay Tep, Soriya Rin, Yoshihisa Uenoyama, Shuichi Matsuyama, Satoshi Ohkura, Tetsuma Murase, Mitsuo Nunome, Yasuhiro Morita

**Affiliations:** 1Asian Satellite Campus in Cambodia, Nagoya University, c/o Royal University of Agriculture, Phnom Penh P.O. Box 2696, Cambodia; 2Department of Animal Health and Veterinary Public Health, General Directorate of Animal Health and Production, Phnom Penh P.O. Box 12352, Cambodia; 3Laboratory of Animal Reproduction, Graduate School of Bioagricultural Sciences, Nagoya University, Nagoya, 464-8601, Japan; 4Laboratory of Animal Production Science, Graduate School of Bioagricultural Sciences, Nagoya University, Nagoya, 464-8601, Japan; 5Laboratory of Veterinary Theriogenology, Joint Department of Veterinary Medicine, Faculty of Applied Biological Sciences, Gifu University, Gifu 501-1193, Japan; 6Laboratory of Animal Genetics, Department of Zoology, Faculty of Science, Okayama University of Science, Okayama 700-0005, Japan; 7Asian Satellite Campuses Institute, Nagoya University, Nagoya 464-8601, Japan; 8Department of Clinical Veterinary Sciences, Obihiro University of Agriculture and Veterinary Medicine, Obihiro 080-8555, Japan

**Keywords:** Cambodia, crossbred dairy cow, GRAS-Di, genetic diversity, genome-wide analysis

## Abstract

**Simple Summary:**

The Cambodian dairy sector collapsed due to the civil war. Thereby, the Cambodia dairy sector has not obtained basic information, such as cattle breed and milk production. To encourage the recovery of the Cambodian dairy sector, the present study aimed to reveal the genetic variation and the milk production in Cambodian crossbred dairy cattle. Initially, we conducted interviews to understand the breeding background and milk production of two local farms (Farm R and Farm M) in Cambodia. The percentage (%) of milk fat content in Farm R was higher than that in Farm M. A genome-wide analysis of the genetic characterization for 75 cows in the two dairy farms implies that some cows in Farm R are genetically far from other crossbred cattle and retain a higher proportion of genetic background derived from Cambodian-native cattle. The present study indicated genetic characteristics and milk composition in Cambodian crossbred cattle. Gene character in Cambodian local crossbred cattle could contribute to milk production in the Cambodian dairy system. Thus, our basic genetic study could provide a new breeding strategy by using local Cambodian crossbred dairy cattle to establish an adequate dairy strain.

**Abstract:**

To improve the dairy sector in Cambodia in the future, we aimed to reveal the genetic variation and the milk production in Cambodian crossbred dairy cattle. We calculated the percent (%) milk fat content and the average milk yield per cow (L/day) for two farms (Farm R and M) based on the farmers’ records and interviews. The crossbred cows originated from Cambodian local farmers and Thailand breeders in Farm R, whereas the crossbred cows originated in Thailand breeders in Farm M. Then, we performed genetic characterization for 75 individuals from the two farms and an individual Japanese pure Holstein-Friesian cow based on 133,705 single nucleotide polymorphisms (SNPs) obtained by the GRAS-Di method. The milk fat contents in the bulk milk in the dry season and the average milk yield per cow on Farm R were 3.77 ± 0.98% and 7.81 ± 2.66 L/day, respectively, and were higher than those on Farm M (3.35 ± 0.54% and 6.5–7.5 L/day). Cattle originating in Cambodia in Farm R possessed a unique genetic character different from cattle from Thailand in Farm M. The present study suggests that the differences in milk fat content between the two farms might be explained by the genetic differences in crossbred cows.

## 1. Introduction

Historically, Cambodia produced enough milk for domestic consumption, and exported it to neighboring countries, such as Thailand and Vietnam, in Southeast Asia, before the civil war between the 1970s and 1990s. However, the livestock sector, including the dairy industry, was destroyed by the civil war [1]. Unfortunately, once the dairy farming system collapsed, knowledge of cattle farming disappeared in Cambodia. After the civil war, cattle husbandry was a very common practice for rural small-scale farmers, although cattle raising was not for production [1]. Instead, in this period, cattle were used for draft power for tillage and transportation. Cattle waste served as raw material for fertilizer and biogas power plants [2], and the cattle themselves were a stock of wealth for conversion into cash for household expenses [3]. Currently, there are around 16 million potential consumers of milk products in Cambodia. Domestic demand of milk and dairy products has increased rapidly and 34,053 kilotons was traded in 2013, which is 28.7% more than the previous year, according to the UN commodity trade statistic database (FAOSTAT, 2013). However, almost all dairy products, including milk, were imported, and the imports increased from 4 million USD in 2009 to 15 million USD in 2013 (FAOSTAT, 2013). Therefore, the dairy industry is a promising industry in Cambodia and is also important for national health and economic growth.

Traditionally, two cattle species have been recognized, *Bos taurus* (humpless cattle) and *Bos indicus* (zebu cattle), in tropical regions. There is no reproductive barrier between them [4]. Both species, as well as the crossbred cattle, are raised in farms in Cambodia (Cattle Policy Report, Cambodian MAFF, 2014). Three cattle breeds, the Kor Khmer (or Khmer), the Hariana, and the Brahman, are well known in Cambodia. The Kor Khmer (or Khmer, *Bos indicus*) is a domesticated breed, which has small hump and yellow hair, and originated from the wild Banteng (*Bos javanicus*) [5], which is classified as an endangered species by the International Union for Conservation of Nature (The IUCN Red List of Threatened Species 2016); the Hariana (*Bos indicus*) is an exotic breed, imported from India for draft power in 1956; and the Brahman (*Bos indicus*) were imported from the Philippines, as a donation from Lutheran Christians in 1984 (Cattle Policy Report, MAFF, 2014). Since the 2000s, various breeds have been introduced to Cambodian cattle farmers, with the spread of artificial insemination techniques. For example, in 2002, Heifer International, a non-government organization in the United States (https://www.heifer.org/, accessed on 1 December 2021), imported and introduced the Jersey breed semen to the farmers in Koh Krabei village, Prek Tmey Commune, Kandal Province, and thus, crossbred cattle were developed locally. In 2004, the Santa Gertrudis semen and Barbara semen were imported from the United States, under the EU project, and introduced in Takeo, Kampong Speu, and Pursat provinces. The Cambodian Center for Study and Development imported the Holstein-Friesian-Brahman, crossbred cattle from Thailand, in 2011. Cambodia had temporary opportunities to access several types of bull semen, but the genetic variations in imported semen were not diverse, because only a few institutes and communities were assigned to these breeding projects. Furthermore, cattle breeding was conducted without any integrated strategies, across the local communities. Therefore, the breeds for production, particularly dairy, have not been strategically managed by the Cambodian cattle industry so far. Elucidation of the genetic background of dairy cattle can help improve breeding practices, which can aid in the development of the dairy industry in Cambodia.

During the last decade, several cost-effective approaches have been developed for analyzing genome-wide single nucleotide polymorphisms (SNPs) in many individuals with high genetic diversity, including restriction site-associated DNA sequencing (RAD-seq and ddRAD-seq) [6,7], and multiplexed inter-simple sequence repeat genotyping by sequencing (MIG-seq) [8]. Recently, attention has been paid to genotyping by random amplicon sequencing-direct (GRAS-Di) [9], as another cost-effective genome-wide genotyping method [10,11]. GRAS-Di performs genotyping for PCR amplicons which are retrieved throughout the genome using random primers. GRAS-Di produces fewer missing data than RAD-seq [10] and more SNPs than MIG-seq [11]. Thus, GRAS-Di has been used widely, for example, to assess the population genetic structure of mangrove fish [9], the Tsushima leopard cat [12], and small apes [13].

As a tentative assessment of the genetic background of Cambodian dairy cattle, we conducted a genome-wide survey of two dairy farm populations, using GRAS-Di analysis, in the suburb area of Phnom Penh city, Cambodia. We then performed STRUCTURE analyses of the two farm populations to understand their genetic characteristics, in comparison with a pure Japanese Holstein-Friesian cow, as a reference. In this study, we aimed to reveal the genetic variation and the milk production in Cambodian crossbred dairy cattle and provide information for improving the efficiency of future cattle crossbreeding strategies to establish an adequate dairy strain

## 2. Materials and Methods

### 2.1. Sample and DNA Extraction

This study was conducted with permission from the Committee of the Care and Use of Experimental Animals at the Graduate School of Bioagricultural Sciences, Nagoya University (Approval number AGR2020061), General Directorate of Animal Health and Production (GDAHP), and MAFF in Cambodia (Approval number 2489 from GDAHP).

Initially, we conducted interviews to understand the breeding background, cattle supply, and milk production for two local farms (Farm R and Farm M), both of which are located in Cambodia. The herd sizes of Farm R and Farm M were 35 and 300 individuals, respectively. The yearly averages milk yield per cow (L/day) were obtained from the farmers’ records and interviews. The milk samples were collected randomly in dry season from November to March, and 92 and 34 samples were obtained from Farm R and Farm M, respectively. The percent (%) milk fat content was measured using FT-IR analytical equipment (MilkoScan Mars, FOSS, Hilleroed, Denmark) in our laboratory in Cambodia.

Hair roots were collected from 75 crossbred cows at the two farms, from all lactating cows in Farm R and randomly from lactating cows in Farm M; in addition, from one pure Japanese Holstein-Friesian cow at a Japanese dairy farm (Nagoya, Japan). At Farm R, a total of 25 crossbred dairy cows were examined, which consisted of 11 Cambodian and 14 Thai cow-derived individuals, and a total of 50 Thai cow-derived individuals were also examined at Farm M. All hair root samples were stored in a deep freezer at −80 °C until DNA extraction, and 40–50 follicles (around 4.5 mg) per animal were used for DNA extraction using the QIAGEN DNeasy Blood and Tissue Kit (Qiagen, Hilden, Germany) according to the manufacturer’s instructions. DNA concentrations were measured using a spectrophotometer (Eppendorf, Hamburg, Germany).

The GRAS-Di and subsequent SNP call analyses were conducted by GeneBay in Yokohama, Japan. The random amplicons were produced with 64 primers and sequenced on a HiSeq4000 sequencer (Illumina, Inc., San Diego, CA, USA), with a paired-end (PE) read length of 150 bp. Low-quality bases and Illumina sequencing adapters were trimmed using the Cutadapt v 2.5 [14].

### 2.2. Analysis of Milk Fat Content Genetic Structure of Crossbred Cows

The fat content (%) in dry season in two farms were analyzed by the Mann–Whitney U test using algorithms in R version 3.5.1 (http://www.R-project.org/, accessed on 30 April 2022), stats R packages. The sites were filtered using the criteria of minor allele frequency (--maf 0.02), missing data (--max-missing 0.2), minimum distance of 100 bp (--thin 100), and Hardy–Weinberg equilibrium (--hwe 0.05) using VCFtools [15]. A total of 133,705 SNPs were obtained from the current genetic survey and used for further analyses. The filtered VCF file was converted to the PLINK format using Tassel 5 [16], and the output files were converted to the STRUCTURE format using PLINK1.9 [17]. A principal component analysis (PCA) was carried out using the TASSEL 5 [16]. The SNP data were analyzed using the program STRUCTURE 2.3 [18] to estimate the genetic differences between cows in the two farms. The numbers of assumed genetic groups (K) were set from one to five. In this study, analyses were performed in five iterations for each K with sampling periods of 100,000 Markov chain Monte Carlo (MCMC) generations after burn-in periods of 100,000 generations. The admixture model and the correlated allele frequency model [19] were applied. The optimal number of genetic groups was identified using the Evanno method [20] implemented in the program STRUCTURE HARVESTER [21]. The results of the five iterations at each K were averaged and visualized using CLUMPAK [22].

## 3. Results

### 3.1. Summary of Milk Production in the Farms

The Farm R cattle had different breeding backgrounds in the herd. This farm had two types of cattle: a Cambodian crossbreed between a Holstein-Friesian cow from Thailand and a Cambodian-native sire (zebu type, Kor Khmer); and an imported Thai crossbreed between a Thailand Holstein-Friesian and a Thailand Brahman sire. These crossbreeds were developed by local Cambodian farmers. In contrast, the Farm M cows were all crossbred between Holstein-Friesian (from New Zealand) and Brahman breeds (Table 1). There was no pedigree relationship between these cow populations. The milk fat contents in the bulk milk on Farm R were higher than those from Farm M, 3.77 ± 0.98 % vs. 3.35 ± 0.54 %, *p* < 0.01, Mann–Whitney U test. The milk yields based on interviews with farmers on Farm M were 6.5–7.5 L/day, and those were different from the average of milk yields of Farm R (7.81 ± 2.66 L/day).

### 3.2. Genetic Structure

Figure 1 shows the first two principal components in PCA. The first PC accounts for 4.5% of the underlying variation and the second PC condenses 2.6% of the variation. Four individuals of Cambodian-originated hybrids in Farm R and five individuals from Farm M were placed to the left of the other individuals. Besides, one individual from each of Brahman-originated hybrids in Farm R and Farm M were distantly placed above the other individuals (Figure 1). The STRUCTURE HARVESTER exhibited the highest and second-highest Delta K at K = 2 and K = 3, respectively, suggesting that the examined cattle could be assigned to two or three genetic clusters (Figure 2A). The probability of each individual assigned to two (K = 2) and three (K = 3) genetic clusters was determined (Figure 2B). At K = 2, one pure Holstein-Friesian cow (N256) showed a high proportion of one genetic cluster, indicated by blue. Almost all individuals examined in this study showed the blue cluster in high proportions, indicating the high proportions of Holstein-Friesian-derived genetic characters. Meanwhile, the proportion of orange clusters, which could indicate a genetic cluster of Asian cattle breeds other than Holstein-Friesian, was quite low compared with the blue clusters, except for some individuals from Farm R. The assignment probability of the blue genetic cluster in each individual did not differ between K = 2 and K = 3, while assignment probability of the orange cluster at K = 2 was replaced in some individuals by the third genetic cluster shown in purple at K = 3. In particular, relatively high proportions of the purple cluster were observed in 4 of 11 crossbred Holstein-Friesian and Cambodian native cattle in Farm R. On the other hand, crossbred cattle from Thailand, in both the farms, had a low proportion of the purple cluster: 0 of 14 in Farm R, and 4 of 50 in Farm M.

## 4. Discussion

We first obtained a large number of SNPs, using the GRAS-Di analysis, from the genetic surveys of crossbred dairy cows in Cambodia. The present study implies that the difference in milk fat content between the two farms was explained by the genetic differences in each crossbred cow, which was originated from other regions. It might be explained that local Cambodian cattle (other than dairy, used for draft power for tillage and/or transportation) have different genetic characteristics from the Thai breeds (of the Brahman breed). The present study had limitations in the samples such as a low number of collection sites; therefore, further studies are needed.

### 4.1. Genetic Characteristics of Cambodian Dairy Cattle

Local farmers in Cambodia could face some limitations in purchasing cattle semen due to a lack of knowledge and skills for artificial insemination and difficulties in accessing semen markets. All semen from dairy bulls is imported from neighboring countries at a high price [23]. Interviews with local Cambodian farmers, at each farm, revealed that their dairy crossbreeding methods were mostly random and involved local sires. In addition, for cows’ pedigree information, we could not track each cow’s history due to the lack of a traceability system in Cambodia, and we revealed only that each farm purchased their cows from different breeders, who fed crossbred cattle. This suggests that the Cambodian dairy cattle are genetically far from the purebred Holstein-Friesian breeds. However, our genetic analysis of the populations, based on genome-wide SNPs, revealed a high genetic similarity between Cambodian dairy cattle and the Japanese Holstein-Friesian breed, as obvious in the high proportions of the blue clusters in both the Japanese dairy and the Cambodian cattle. We consider the blue and orange genetic clusters at K = 2 in the STRUCTURE plots to be representative of the *Bos taurus* (humpless cattle) and *Bos indicus* (zebu cattle), respectively. This suggests that the local Cambodian dairy farmers cannot maintain their pedigrees by using pure semen from dairy breeds, leading to the restricted number of individuals with the high proportion of indicus-type (orange) genetic clusters at K = 2. The indicus-type genetic clusters at K = 2 were subdivided into two different clusters (orange and purple) in some crossbred individuals between the Holstein-Friesian and Cambodian native cattle at K = 3. The individuals that showed relatively higher proportions of the purple cluster at K = 3 were individuals that were placed to the left in the PCA plot, in addition, four individuals were crossbred between Cambodian-native and Holstein-Friesian. Therefore, it was implied that the genetic character included in the purple cluster is explained by Cambodian-native cattle. On the other hand, as to the two individuals that were placed above in the PCA plot, even though we could not find obvious differences to other individuals in the STRUCTURE plots, they tended to have a little bit higher proportions of the orange cluster than others. Further studies are needed to investigate whether the orange cluster was a genetic cluster of zebu cattle. A previous study suggested that several *Bos* species were involved in the history of Asian cattle [4]. Our result implied that the purple genetic cluster at K = 3 represents the wild Banteng cattle, which is a native species of Southeast Asia. Cambodian local cattle, Kor Khmer is a domesticated breed of the wild Banteng (*Bos javanicus*). Cows that had a large portion of the purple cluster in Farm R were crossbred between local Cambodian cows and a purebred Holstein-Friesian sire. On the other hand, some cows on Farm M also had a large proportion of the purple cluster. Wild Banteng cattle have been domesticated in several regions in Southeast Asia [24], the Brahman breed has originated in local zebu cattle in India and the Thai Brahman breed has developed originally in Thailand [25], suggesting Brahman sires developed in Thailand might have the same genetic characteristics as Cambodian native cattle, Kor Khmer. Our genetic survey using 133,705 SNPs suggested a much higher genetic contribution of the Holstein-Friesian breed to the genetic backgrounds of the two farm cattle populations. Although previous studies, in which 30 Cambodian native cattle were genotyped using 58 [26] and 117 SNPs [27], revealed genetic differences between the Cambodian native cattle and *Bos taurus*-derived populations, the genetic erosion of the Holstein-Friesian breed might have occurred locally, in dairy cattle populations from Cambodia. In this experiment, we could not obtain enough samples to conduct an association analysis; therefore, further experiments are needed. However, our genetic survey also revealed slight differences in the genetic characteristics between farms R and M. Farm R appeared to retain a higher proportion of genetic background derived from the local, Cambodian-specific cattle, than Farm M.

### 4.2. Implications for Future Breeding in Cambodia

The study indicated differences between the two farms in their relationship between milk yield and milk fat content. It is speculated that the local Cambodian cattle have effective genetic diversity for maintaining milk fat content high under tropical conditions. Generally, the production of milk, and its fat and protein contents, significantly decline under tropical conditions [28,29]; however, the previous study suggested that the genetic character could contribute to maintain milk components high in the tropical region [30]. In addition, nutrient affects milk fat content [31]. In the present study, the data of milk fat content was obtained from the bulk, and 11 of 25 lactating cows on Farm R were crossbred between Cambodian native and Brahman. The climate conditions in the two farms were not significantly different, because both farms are located just across the Mekong River. Moreover, Farm M feeds rich nutrient silage, which contributes to improvement of milk components, to cows compared with Farm R. Whereas, the percentage (%) of milk fat content in Farm R was higher than that in Farm M. In the experiment, we observed genetic data by GRAS-Di method. Unfortunately, the obtained genotype data did not cover all SNPs throughout cattle genome and did not include genes that were reported to have a positive relationship to milk yield and compositions under tropical condition. Although we have understood further experiments are needed, it is speculated that the local Cambodian cattle used in crossbreeding in Farm R have effective genetic diversity for maintaining milk fat content high under tropical conditions.

This possibility could support future cattle breeding programs in Cambodia. Generally, the fat content of milk is one of the biggest concerns for the dairy industry in the tropics. Previous reports indicate that many genetic characteristics support milk yield and milk quality, under heat stress conditions, and many dairy sectors consider improving their dairy strain based on this genetic information in tropical areas [32]. In Cambodia, almost all dairy cattle are not purebred and without pedigree certificates. Therefore, understanding the genetic character of existing crossbred dairy cattle is important for improving the efficiency of future cattle breeding. The present results could provide a new strategy, such as using local Cambodian cattle breeds to establish an adequate dairy strain. For example, a specific dairy crossbreed, made from local zebu and commercial dairy cattle, was created in Brazil, South America [33].

## 5. Conclusions

This report is the first to perform genetic characterizations of Cambodian crossbred dairy cows using a genome-wide analysis tool. Although further experiments are needed, the present study provides useful information for Cambodian cattle breeding stakeholders. In addition, our findings could be used to improve dairy breeding programs in Southeast Asian countries interested in increasing milk production in the tropics.

## Figures and Tables

**Figure 1 animals-12-02072-f001:**
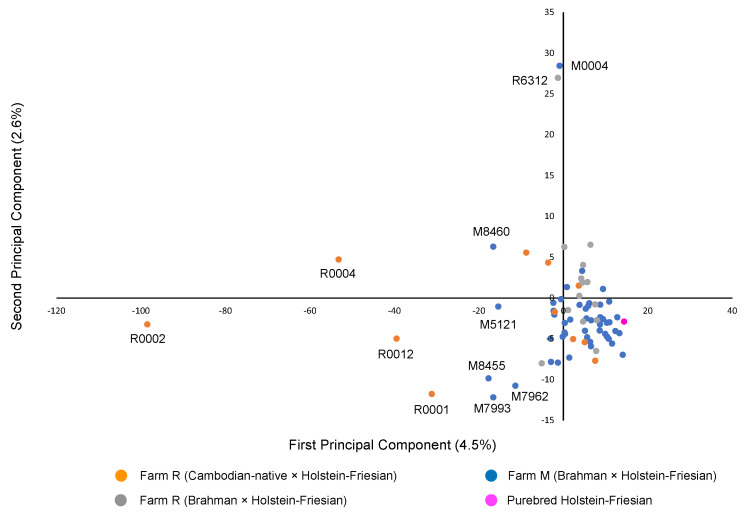
PCA of the first two principal components. Individuals of crossbreeds between Cambodian-native cattle and Holstein-Friesian in Farm R, individuals of crossbreeds between Brahman and Holstein-Friesian in Farm R, individuals from Farm M, and the pure Japanese Holstein-Friesian individual are indicated with orange, grey, blue, and magenta-filled circles, respectively. Eleven individuals that are distantly placed from the other individuals are shown with their ID.

**Figure 2 animals-12-02072-f002:**
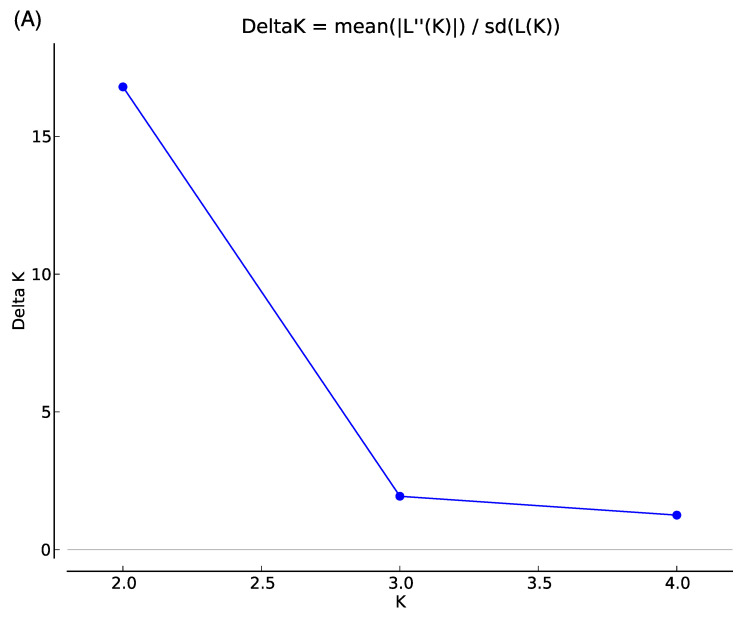
Delta K values at K = 2 to K = 4, generated by STRUCTURE HARVESTER. (**A**) STRUCTURE plots for the probability of assignment to two (K = 2) and three (K = 3) genetic clusters, for the pure Japanese Holstein-Friesian breed, and 75 Cambodian dairy cattle, from two different farms (**B**). Each horizontal bar indicates an individual, and each color indicates the probability of assignment to one of the genetic clusters.

**Table 1 animals-12-02072-t001:** Summary of the cattle breeds and milk production in two farms.

		Cattle Breed				Milk Production	
	Location (Province)	Herd Size	Cow Provider	Breeding	Breeding Method	Average Amount/Cow (Mean ± SD)	Average Milk Fat (Mean ± SD)
Farm R	Phnom Penh city	35	Cambodian farmers	Holstein-Friesian (Thailand) × Cambodian Local	AI (Local, mating)	7.81 ± 2.66 L/day	3.77 ± 0.98% **
			Thailand breeders	Holstein-Friesian (Thailand) × Brahman	AI		
Farm M	Kandal	300	Thailand breeders	Holstein-Friesian (New Zealand) × Brahman	AI	6.5–7.5 L/day ^†^	3.35 ± 0.54%

** Indicates a significant difference, Mann–Whitney U test, *p* < 0.01. ^†^ Based on interviews with farmers.

## Data Availability

Not applicable.

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
