# Peer review of "A Survey of Genome-Wide Genetic Characterizations of Crossbred Dairy Cattle in Local Farms in Cambodia"

_animals, 2022, doi:10.3390/ani12162072_

Round 1
Reviewer 1 Report
Thank you for including the PCA plot, although more discussion would be appreciated. It appears that PC1 separates Cambodian (Banteng?) content from Holstein, which agrees with the K=3 STRUCTURE plot - samples with the highest content of the third (purple) cluster have the most negative values for PC1. PC2 is is harder to interpret - looks like it may separate Brahman from Holstein, but agreement with STRUCTURE isn't as clear. It is interesting that most cows cluster near the purebred Holstein in the PCA plot, and have high Holstein content in STRUCTURE plots.
It is difficult to attribute the difference in milk fat content between the farms solely to genetic differences between the farms. The apparent difference in diets and other factors may contribute to that difference. If available, records from the Cambodian- and Thai-based cattle on farm R might be a better comparison with different genetics on the same farm, although larger N may be needed for a meaningful comparison.
lines 171-174, Table 1) what are values following the +/- signs? Presumably standard deviations which would translate to standard errors of 0.09 to 0.10?
Author Response
- Thank you for including the PCA plot, although more discussion would be appreciated. It appears that PC1 separates Cambodian (Banteng?) content from Holstein, which agrees with the K=3 STRUCTURE plot - samples with the highest content of the third (purple) cluster have the most negative values for PC1. PC2 is is harder to interpret - looks like it may separate Brahman from Holstein, but agreement with STRUCTURE isn't as clear. It is interesting that most cows cluster near the purebred Holstein in the PCA plot, and have high Holstein content in STRUCTURE plots.
Our response
we really appreciate your constructive suggestion. Individuals that showed higher values in PC2 tended to have slightly higher proportions of the orange cluster, but it was not so obvious. We have added a discussion in the revised manuscript (L255-L258).
- It is difficult to attribute the difference in milk fat content between the farms solely to genetic differences between the farms. The apparent difference in diets and other factors may contribute to that difference. If available, records from the Cambodian- and Thai-based cattle on farm R might be a better comparison with different genetics on the same farm, although larger N may be needed for a meaningful comparison.
Our response
We appreciate your pertinent observations and suggestions. I agree with you, we need further studies to indicate the relationships between genetics and milk compositions, because we assigned a low number of collection sites to the study. We have revised the manuscript and added the sentences following your suggestions (L230-231).
- lines 171-174, Table 1) what are values following the +/- signs? Presumably standard deviations which would translate to standard errors of 0.09 to 0.10?
Our response
We appreciate your pertinent observations; I revised the sub-headings and add the standard deviations in the table.

Reviewer 2 Report
I re-evaluate the manuscript, and suggest an acceptance because of the conditions of Cambodia.
Author Response
- I reevaluate the manuscript, and suggest an acceptance because of the conditions of Cambodia.
Our response
We appreciate your understanding to our research and the study limitation.

Reviewer 3 Report
I have the following issues.
1) Introduction: Please explain about the current numbers, main areas, and future demands of farmers raising crossbred dairy cattle in the Cambodian dairy industry.
2) Cow samples: Are they unrelated? Please describe about the relationships of the cow samples in the materials and methods.
3) Table 1: The authors should show the average milk yield/cow and milk fat content in each of the two crossbred populations on the farm R.
In addition, the authors should statistically test the milk fat content between the two crossbred populations of the farm R.
4) Line 222-224: Both PCA and Structure analysis revealed the two Brahman-originated crossbred populations may be genetically similar each other. Therefore, I fail to understand that a significant difference in the fat content between the two farms results from genetic difference of the crossbreds.
5) Lines 241-256: The authors explained that orange and purple genetic clusters by structure analysis may be zebu and Banteng cattle, respectively. Therefore, R0002 in Fig2B is considered as a crossbred of Banteng and taurine cattle. However, as described in line 167, the authors described the Canbodian native sire is zebu type. I fail to understand the reason why a crossbred of Banteng and taurus (R0002 in Fig2B) was born from a zebu-shaped sire.
6) Please show the significant difference of the chromosomal regions between the three populations by using the analysis such as haplotype-based hapflk. The authors may discuss the QTL of the fat content on the regions.
Author Response
1) Introduction: Please explain about the current numbers, main areas, and future demands of farmers raising crossbred dairy cattle in the Cambodian dairy industry.
Our response
We appreciate your pertinent observations and suggestions. Unfortunately, the Cambodian government doesn’t have an official report about dairy cattle because in Cambodia, only 5 dairy farms have been established officially, and others are small-holders, they produce milk for unofficial local consumption. Therefore, the Cambodian government does not obtain accurate information about the number of raised dairy cows. On the other hand, the demand of milk and dairy products has increased rapidly. We already described in previous version such as “Demand has increased rapidly and was 34,053 kilo tons in 2013, which is 28.7% more than the previous year, according to the UN commodity trade statistic database (FAOSTAT, 2013). The import of dairy products, including milk, increased from 4 million in 2009 to 15 million US dollars in 2013 (FAOSTAT, 2013). Therefore, the dairy industry is a promising industry in Cambodia and is also important for national health and economic growth.” We appreciate your understanding for the condition of Cambodia. We have revised the manuscript for readers accurate understanding (L63-69).
2) Cow samples: Are they unrelated? Please describe about the relationships of the cow samples in the materials and methods.
Our response
We appreciate your pertinent observations and suggestions. We obtained hair root samples from two farms, and there was no pedigree relationship between these cow populations. We have revised the manuscript and added the sentences following your suggestions (L136-137).
3) Table 1: The authors should show the average milk yield/cow and milk fat content in each of the two crossbred populations on the farm R.
In addition, the authors should statistically test the milk fat content between the two crossbred populations of the farm R.
Our response
We appreciate your pertinent observations and suggestions. Unfortunately, Cambodian dairy sectors doesn’t have enough human resources in analysis of dairy products. Before our study, there are no analytical equipment in Cambodia. From this situation, we could not obtain individual data in milk yield and compositions. Therefore, we did not conduct statistical analyses between two crossbred population in Farm R. We appreciate your understanding for the condition of Cambodia.
4) Line 222-224: Both PCA and Structure analysis revealed the two Brahman-originated crossbred populations may be genetically similar each other. Therefore, I fail to understand that a significant difference in the fat content between the two farms results from genetic difference of the crossbreds.
Our response
We appreciate your pertinent observations. It may have confused you, because of our lack of explanations. The data of milk fat was obtained from the bulk, and 11 of 25 lactating cows on farm R were crossbred between Cambodian native and Brahman. Moreover, the climate conditions in the two farms were not significantly different, because both farms are located just across the Mekong River. Farm M feeds rich nutrient silage, which contributes improvement of milk components, to cows compared with Farm R. Therefore, we speculated that the local Cambodian cattle have effective genetic diversity for maintaining milk fat content high under tropical conditions. We have revised the manuscript and added the sentences (L288-290).
5) Lines 241-256: The authors explained that orange and purple genetic clusters by structure analysis may be zebu and Banteng cattle, respectively. Therefore, R0002 in Fig2B is considered as a crossbred of Banteng and taurine cattle. However, as described in line 167, the authors described the Canbodian native sire is zebu type. I fail to understand the reason why a crossbred of Banteng and taurus (R0002 in Fig2B) was born from a zebu-shaped sire.
Our response
We appreciate your pertinent observations. It may have confused you, because of our lack of explanations. Cambodian-native cattle is Kor Khmer. It originated from wild Banteng but has been crossbred with other cattle, and there is no pedigree certificate. The mature Kor Khmer has medium body size, the body length is about 150-170 cm, with a short neck, dewlap not hanging too much, small hump, medium horns with sharp points (22-27 cm), hanging belly, their colors are brown, red-yellow, white or black and the body weight is around 250-350 kg as reported previously (Sath et al., 2008). We have revised the manuscript and added the reference (L74-76, 170, 348-349).
Sath, K., Borin, K. and Preston, T.R. Survey on feed utilization for cattle production in Takeo province. Livestock Res Rural Dev 2008; 20. http://www.lrrd.org/lrrd20/supplement/sath1.htm
6) Please show the significant difference of the chromosomal regions between the three populations by using the analysis such as haplotype-based hapflk. The authors may discuss the QTL of the fat content on the regions.
Our response
We appreciate your pertinent observations and suggestions. We strongly agree with your suggestion, however, in this experiment, we could not obtain enough samples to conduct your suggestion. We already described Cambodia doesn’t have enough instruments and human resources to conduct milk analysis. Therefore, we could not obtain individual data related to milk yield and compositions. Moreover, it is difficult to obtain the other information related to milk production, such as detailed feed condition. In the experiment, we observed genetic data from GRAS-Di. Unfortunately, the obtained data doesn’t contain complete SNPs that are reported to have a positive relationship to milk yield and compositions under tropical condition. We have understood further experiments were needed. However, this manuscript has revealed new genetic information in Cambodian crossbred dairy cattle. Therefore, we believe this manuscript is suitable for publication. This information has been added to L294-298.
Round 2
Reviewer 3 Report
Line 136-137: I would like to know the unrelated samples within each population.
Author Response
Line 136-137: I would like to know the unrelated samples within each population.
Our response
We appreciate your pertinent observations. It may have confused you because of our lack of explanations. I moved the sentence you commented on to the result section and added further discussion in the discussion section. For cows’ pedigree information, we could not track each cow’s history due to the lack of a traceability system in Cambodia, and we revealed only that each farm purchased their cows from different breeders who fed crossbred cattle. We speculated that each breeder has a different pedigree strain of cattle because the location of each breeder was far from each other, and their community was different. We appreciate your understanding of the condition of Cambodia. We have revised the manuscript for readers' accurate understanding (L174-175, 241-244).

This manuscript is a resubmission of an earlier submission. The following is a list of the peer review reports and author responses from that submission.
Round 1
Reviewer 1 Report
1. Please describe the methods of measuring average milk yield and yearly milk fat content used in this study, although the authors used farmers' records. For example, the authors should provide the additional data such as daily milking frequency, parity numbers and lactation period.
2. In Table 1, the authors should compare the average amounts/cow and average milk fat between two populations of the farm R and discuss the results in the text.
3. Please describe how the authors selected the cow samples of the two farms for NGS study.
4. Please provide PCA results using NGS data for genetic diversity among three dairy cattle populations.
5. Lines 226-228: Please provide an evidence of genetic contribution of the wild Banteng cattle.
Reviewer 2 Report
This study has investigated the phenotype (milk yield and quality trait) and genetic characteristics in Cambodian crossbred dairy cattle, and the results could provide understanding of genetic characteristics of Cambodian crossbred dairy cattle and help to develop future breeding strategy for breeding new dairy strain. Some modifications are needed.
1. There is no analysis to reveal the relationship between milk production and the genetic variation, such as association analysis between phenotypes and SNPs.
2. Remove Figure 1.
3. The burn-in number is not normal for MCMC sampling.
4. Add A and B in the Figure 2.
5. Breeding is usually conducted on pure breed, and how to use the crossbred one for breeding?
Reviewer 3 Report
Interesting use of a new approach of genotyping by sequencing to extract useful information from a small set of genotyped cows. Genotyping by sequencing random amplicons revealed Holstein x local Cambodian crossbred cows were genomically different than Holstein x Thai Brahman. Average milk yield and fat content of the farm having Cambodian-cross cows was sufficient to suggest that Cambodian cattle can make a useful contribution to dairy production.
STRUCTURE results might be strengthened by including a few samples of Thai Brahman and local Cambodian cattle that haven't been crossed with Holstein. Those samples should show up as high content of their respective cluster, further confirming apparent content of the crossbred cows. Principal component analyses might also support the STRUCTURE results - are there distinct clusters with some separation between Thai- and Cambodian-cross animals?
Wording throughout the paper is sometimes awkward but understandable. I do not understand the sentence on lines 244-245 "Also, many reports suggest that heat stress affects the components of milk, resulting in based on the genetic differences [27]."